# Crisis Ocean Modelling with a Relocatable Operational Forecasting System and Its Application to the Lakshadweep Sea (Indian Ocean)

**Georgy I. Shapiro** [1,*]**, Jose M. Gonzalez-Ondina** [2,3]**, Mohammed Salim** [2,4] 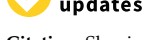**, Jiada Tu** [2] **and Muhammad Asif** [2]

1   School of Biological and Marine Sciences, University of Plymouth, Plymouth PL4 8AA, UK
2   University of Plymouth Enterprise Ltd., Plymouth PL4 8AA, UK
3   Engineering School of Sustainable Infrastructure and Environment, University of Florida, Gainesville, FL 32603, USA
4   School of Ocean Sciences, Bangor University, Bangor LL57 2DG, UK
*   Correspondence: gshapiro@plymouth.ac.uk

**Abstract:** This study presents the Relocatable Operational Ocean Model (ReOMo), which can be used as a Crisis Ocean Modelling System in any region of the global ocean that is free from ice. ReOMo can be quickly nested into an existing coarser resolution (parent) model. The core components of ReOMo are the NEMO hydrodynamic model and Rose-Cylc workflow management software. The principal innovative feature of ReOMo is the use of the Nesting with Data Assimilation (NDA) algorithm, which is based on the model-to-model assimilation technique. The NDA utilises the full 3D set of field variables from the parent model rather than just the 2D boundary conditions. Therefore, ReOMo becomes physically aware of observations that have been assimilated and dynamically balanced in the external model. The NDA also reduces the spatial phase shift of ocean features known as the 'double penalty effect'. In this study, ReOMo was implemented for the Lakshadweep Sea in the Indian Ocean at 1/20°, 1/60°, or 1/120° resolution with and without model-to-model data assimilation. ReOMo is computationally efficient, and it was validated against a number of observational data sets to show good skills with an additional benefit of having better resolution than the parent model.

**Keywords:** ocean modelling; Indian Ocean; data assimilation; downscaling; operational forecast

## 1. Introduction

The need for fine-resolution oceanographic data is growing. These data are used to support fisheries and navigation, and to assess the impact of hazards such as marine litter and contaminants. The accuracy and efficiency of oceanic numerical models have greatly improved in the last decades and they are commonly used operationally, i.e., for real-time forecasts of the ocean state to support and sometimes to replace observations in a move to create "Digital Twins of The Ocean (DITTO)" [1]. The output of a number of global ocean operational models is publicly accessible, for example, from the EU Copernicus Marine Service [2]. Due to high computational cost, the global models have a relatively coarse resolution. They usually do not include tidally induced currents, which are important in coastal and shelf seas. Regional and local operational models have higher resolution and are capable of resolving important meso- and sub-mesoscale features not revealed by the global models. The capability of finer resolution models to represent smaller-scale processes is counterbalanced by the introduction of greater errors and generating ocean features not in phase with reality. There have been a number of studies assessing relative strengths and weaknesses of finer resolution models compared to their coarser resolution counterparts, e.g., [3,4] and references therein. Sometimes, the finer-resolution models run without data assimilation, having only an indirect link to observation by taking boundary conditions from the coarser data assimilating the 'parent' model, see e.g., [5]. However, in recent research, it was found that 'in the absence of data

assimilation, the high-resolution model is not able to properly reproduce the observed phases of mesoscale structures' [6]. Therefore, the downscaled regional models would benefit from assimilating good quality data.

Being operational, finer-resolution models automatically import necessary external data such as meteorological forcing and ocean observations, carry out quality checks, run the core ocean circulation models, and disseminate the outputs to the end-user. However, such finer resolution models may not exist in a specific area of a marine accident, oil pollution or release of pollutant due to ship collision, or other man-made or natural disaster. The setting up, calibrating, testing, and validating of a new regional model is a long process taking up to a few months. A similar problem exists in numerical weather prediction where a solution is provided by weather forecasting Crisis Area Models (CAMs) [7].

This paper presents a relocatable ocean modelling system (ReOMo), the development of which was inspired by atmospheric CAMs. ReOMo can be quickly set up in a new area as many processes involved in the set up the model are automated. ReOMo uses a computationally efficient Nesting with Data Assimilation (NDA) technique developed in [8]. In contrast to conventional approach where only 2D data sets for creating lateral boundary conditions are used from the external model, the NDA method utilises the full 3D data set produced by the external model within the domain of ReOMo. As many other regional ocean models, see e.g., [9] and references therein, ReOMo is nested into a coarser resolution global model.

ReOMo was set up and tested in the Lakshadweep Sea, which is located in the tropical Indian Ocean to the west of the south-western part of Indian peninsula. The sea provides a vital supply of proteins to the people of the Indian state of Kerala and beyond. It also is subject to monsoon influence, and is highly dynamic with strong mesoscale activity. The paper is organised as follows. Section 2 describes data and methods, Section 3 presents the results, highlighting the positive effect of model-to-model data assimilation, and the discussion is presented in Section 4.

## 2. Data and Methods

The relocatable ocean modelling system contains Nucleus for European Modelling of the Ocean-NEMO version 3.6 [10] as its core modelling engine and Rose-Cylc software [11] for automated workflow management including pre-processing, initialisation, acquiring necessary external data, running the ocean model, data assimilation, and uploading the results to a data storage pool. For this work, ReOMo was implemented in two versions named LD20_DA and LD20_noDA, with and without data assimilation (DA), respectively. Both versions are set in the domain 7.5–14.5° N, 68–78° E and have horizontal resolution of $1/20°$ and 50 depth levels. This paper also briefly shows some results from ReOMo being applied to higher resolution models LD60 at $1/60°$ (over the same domain) and LD120 at $1/120°$ within the domain of 8.5–11° N, 74.5–77° E.

### 2.1. NEMO Model

The NEMO model has been extensively tested in multiple studies covering coastal, regional, and global domains, and it has shown good forecasting skills [12] subject to proper combination of simulation options and tuning parameters. The NEMO model is set on the Arakawa C-grid, and uses the variable volume non-linear free surface and the Total Variation Diminishing time-stepping scheme. Diffusion and viscosity coefficients were taken from [13] and further adjusted by calibrating the model outputs against observations. Both versions of LD20 use the Laplacian formulation of the Smagorinsky scheme for horizontal viscosity, and a combination of Laplacian and bi-Laplacian operators for horizontal diffusivity. Vertical diffusion and viscosity coefficients are provided by the General Length Scale (GLS) turbulence closure scheme using the k-ε option. Both versions of LD20 model (with and without data assimilation) used the following user adjustable parameters: the baroclinic and barotropic time steps of 120 and 6 s respectively, compilation keys: *key_iomput, key_vectopt_loop, key_zco, key_dynspg_ts, key_ldfslp, key_mpp_mpi,*

*key_bdy*, *key_tide*, *key_check_nan*, *key_zdfgls*, *key_vectopt_loop*, *key_dynldf_smag*, *key_dynldf_c3d*, *key_traldf_smag*, *key_traldf_c3d*, *key_orlanski_npo_imp*.

The model is forced by wind velocities and air temperature obtained at 10 m above surface, total downward shortwave radiation flux, total longwave radiation flux, precipitation, and relative humidity. The wind stress and surface radiation fluxes are estimated using the bulk formula of Large and Yeager [2]. The flow relaxation scheme (FRS) is applied at the open boundary using an unstructured NEMO BDY algorithm [12] on a 10-node-wide sponge layer for temperature, salinity, and baroclinic velocity, while the barotropic velocities are treated using the usual Flather radiation conditions [10]. Tidal currents were added to the barotropic velocities obtained from the external model, by assuming superposition. The domain of the LD20, LD60, and LD120 models has 50 geopotential depth levels between 0.5 and 4600 m. The number and type of computational levels (z-level, sigma, and multi-enveloping s-coordinates) are user-selectable in the usual way [14]. Both LD20 models output 3-hourly instantaneous and daily average values for temperature, salinity, 3D velocity, and sea surface height.

### 2.2. External Data

Global bathymetry is taken from GEBCO database [15] at 1 arc minute resolution and linearly interpolated to the model horizontal grid. The meteorological forcing is provided by the global atmospheric model run by the UK Met Office [16] at 3-hourly intervals for most variables and 1-hourly intervals for winds. Initial and open boundary conditions for potential temperature, salinity, and meridional and zonal velocities, are taken from a global model at 1/12° resolution with 50 geopotential depth levels available as GLOBAL_REANALYSIS_PHY_001_030-TDS via EU Copernicus Marine Service [2]. Nine tidal components: M2, S2, K1, O1, Q1, P1, N2, K2, M4 are taken from the Topex-Poseidon global tidal model (TPXO) version 7.1 [17].

Sea surface temperature (SST) for model validation is taken from the Operational Sea Surface Temperature and Ice Analysis (OSTIA) at 1/20° resolution available via the UK Met Office website [18] and the GHRSST-MUR Level 4 Group for High Resolution Sea Surface Temperature-Multiscale Ultrahigh Resolution data set (hereafter called GHR-MUR) available via the GHRSST website [19]. Vertical profiles of temperature and salinity for model validation are taken from Argo float observations available from [20].

### 2.3. Data Assimilation

The LD20_DA model utilises a Nesting with Data Assimilation (NDA) methodology [8]. This method assimilates a 3D output from parent global data, which is a data assimilating model in its own right and thus it links, however indirectly, the child model with real-world observations. For consistency, ReOMo uses the same global model [2] for both data assimilation and for providing boundary conditions. The benefit of assimilating data from a good quality coarser model rather than directly from observations is that the model data are on a regular grid in space and time and the NDA method is computationally very efficient. The principal difference with common data assimilation schemes is in the calculation of error covariance matrices (ECMs), which are required to minimise the cost function in order to get the best possible estimate of the true field [21,22]. In common DA methods, the background (model) and observational errors are assumed to be uncorrelated [23–25]. It is further assumed that the observational errors are spatially uncorrelated between them so that the observational ECM is diagonal; see, e.g., [26]. The latter assumption is unlikely to be true in case of using parent model data instead of observations and a different approach to estimating ECMs is required. The details of assessing the background and 'observational' ECMs in case of model-to-model data assimilation are given in [8].

The NDA process contains two components. The first component includes Stochastic-Deterministic Downscaling (SDD) which projects the outputs of the coarse parent global model onto the finer resolution nested child model grid [27]. The SDD is based on the concept of objective analysis; it represents the best linear unbiased estimator (BLUE), see [28],

and is capable of recovering finer-scale details which are only embryonically revealed by the parent model. The second component includes assimilation of the downscaled 3D parent model data into the child model. The SDD includes (i) calculations of correlation functions and objective analysis weights [29] which are carried out only once at the start of ReOMo operation, and (ii) the application of these weights to perform SDD repeatedly at each DA cycle. The second component includes calculation and application of Kalman filter gain weights at each DA cycle, as detailed below. In the case of LD20_DA, the DA cycle is five days and assimilation occurs at 00:00 GMT. The length of the DA cycle is selected based on sensitivity tests, and can be easily changed by the user if required.

The SDD requires the knowledge of correlation functions for fluctuations (deviations from statistical mean) of each field variable at each 3D grid point of the finer model. In ReOMo, statistical means and fluctuations are estimated using the parent coarse model data and applying the ergodic hypothesis in that the ensemble averages are replaced by time averaging. The correlation functions are calculated using a two-length scale technique, i.e., by fitting the spatial correlations to the sum of two Gaussian functions with different dispersions [30]. As the SDD method is concerned with the recovery of fine-scale variations, only the curve with the short length scale is used for further processing. In the case of LD20_DA, the correlation lengths (CorLen) vary horizontally in the range of 15 to 85 km; however, they show only minor variability with depth and between the variables, a fact consistent with previous studies [31,32]. Therefore, the values of the CorLen calculated for the sea surface temperature at each external model node are used for all variables. The next step is to linearly interpolate the CorLen from the external onto the finer nested model grid. The calculation of the CorLen for the SDD method is different from calculations of Error Covariance Matrices (ECMs) used in many DA methods [23]: the CorLen matrix is based on correlation of the fluctuations of field variables themselves which are known with some accuracy, while the ECMs require the knowledge of model and observational errors, which are not normally known, and hence various proxy methods are used [33]. In the case of LD20_DA, the total number of SDD weights for four variables: T, S, U, and V is 505,447,676.

The following additional steps are carried out at each assimilation cycle in case the computational depth levels of the child and parent models do not coincide, which happens in the case of LD20_DA. The parent model data are vertically interpolated onto the LD20 depth levels but keeping the original (coarse) horizontal resolution. This step takes place before the SDD and may require some simple horizontal extrapolation to generate parent model data 'under the seabed' when bathymetries for the parent and child models do not match.

The CMEMS global model provides daily average data while instantaneous data at midnight is required for DA. Therefore, the external model data are shifted to 00Z hours by applying a 'midnight correction'. Its value is calculated by subtracting the daily average and instantaneous (at 00Z) data at each 3D LD20 grid node and assuming that the day-to-night difference is the same for both models.

At this point, ReOMo has two data sets at the same (fine) 3D grid and at the same time. The final step is to combine these data sets using the NDA technique by applying a zero-dimensional Kalman filter to fluctuations at each child model grid node to reduce RMSE and replacing the statistical mean from the nested model with that of the external model to reduce bias. A brief mathematical description of the equations used to obtain the analysis state is given below.

Let us consider a cost function $J_S$:

$$J_S(x') = \left(x' - x'^b\right)^T \mathbf{B}^{-1} \left(x' - x'^b\right) + \left(S'(y) - x'\right)^T \mathbf{R}^{-1} \left(S'(y) - x'\right)$$

where $x$, $x^b$, and $y$ denote vectors of the true state, fine, and coarse model respectively; the primed variables denote deviations from statistical means, $\mathbf{B}$ and $\mathbf{R}$ are ECMs for the nested and parent models, and $S\prime$ is the projection operator, which was used at the SDD step of data assimilation. The minimum of the cost function is achieved if:

$$2\left(x'^a - x'^b\right)^T \mathbf{B}^{-1} - 2\left(S\prime(y) - x'^a\right)^T \mathbf{R}^{-1} = \mathbf{0}$$

where $x'^a$ is the deviation of the analysis vector from its statistical mean. Let us introduce the error correlation matrices $\mathbf{C}_B$ and $\mathbf{C}_R$ for the child and downscaled parent (after the SDD step) models:

$$\mathbf{B} = \mathbf{V}_B\mathbf{C}_B, \quad \mathbf{R} = \mathbf{V}_R\mathbf{C}_R$$

where $\mathbf{V}_B$, $\mathbf{V}_R$ are diagonal matrices containing the respective error variances at each fine-grid node. After some algebraic manipulation, one can obtain the following expression for each element $x_i^a$ of the state vector $\mathbf{x}^a = x_i^a = Kw_i x'^b_i + (1 - Kw_i)S\prime(\mathbf{y})_i + \langle S(\mathbf{y})_i \rangle$. where $Kw_i$ is the Kalman gain coefficient at each fine-grid node:

$$Kw_i = \frac{V_{Rii}}{V_{Rii} + V_{Bii}}$$

Note that the statistical mean $\langle S(\mathbf{y})_i \rangle$ is calculated using only the data from the coarse model. The details of this procedure are presented in [8].

*2.4. Workflow Engine*

ReOMo automates many of the computational steps required for running an operational model in a new area of the ocean. The user is required to select the limits of the domain, horizontal resolution, barotropic and baroclinic time steps, and computational depth levels. The weights for interpolation of meteorological fields are also calculated off-line using standard NEMO tools. ReOMo takes care of many other tasks including compilation of NEMO from a source code using a Flexible Configuration Management build system [34].

For running in the operational model, ReOMo utilises the Rose-Cylc software environment for managing the operational modelling tasks [11]. Cylc is the workflow engine that runs the suites of interdependent jobs. The Rose component acts as a toolkit for editing, writing, and executing the application configurations and provides a helpful graphical user interface. Rose-Cylc is implemented in ReOMo to submit jobs from a controlling Linux computer to the workhorse HPC cluster. For Lakshadweep Sea, both LD20 models use 71 computing cores on an HPC cluster.

Before ReOMo starts its simulations, a few pre-processing tasks have to be completed off-line. These include: (i) creation of operational bathymetry from GEBCO global data set; (ii) generation of 3D computational grid; (iii) creating initial conditions for temperature, salinity, zonal, and meridional components of velocity from the external model or climatology; (iv) calculating interpolation weight files for meteorological data; (v) preparation of tidal boundary conditions (amplitude and phases); and (vi) calculation of downscaling weights required for the SDD step of DA.

ReOMo contains a set of applications; some are run only once at the beginning of simulation, and others are run repeatedly every model day. The run-once tasks include (i) compilation of Nemo executive module from the source code; (ii) copying pre-processed initial conditions into the work space; and (iii) creation of the original namelist for NEMO computations. The daily repeated tasks include (i) updating of the namelist; (ii) interpolation of meteorological forcing onto the model grid; (iii) preparation of data on river discharges if necessary (LD20 does not include rivers); (iv) execution of NEMO module; (v) data assimilation—LD20_DA does it every five days; (vi) creating the restart file for the next day simulations; (vii) transferring the daily average and 3-hourly instantaneous results from NEMO simulation to the external storage system; and (viii) cleaning the operational disk to prepare space for further simulations. The cleaning is carried out with a five-day delay to allow simple automatic or manual re-run of tasks in case of failure. In order to save time, some tasks (e.g., interpolation of meteorological forcing, preparation of river data, lateral boundary conditions for the next day simulation) are carried out concurrently with or before NEMO model runs.

ReOMo generates the initial working namelist required by the NEMO model by using pre-recorded parameters. The end-user can modify these parameters. At the start of simulations, ReOMo runs NEMO from the initial conditions for one day and creates the

first restart file for further repeated cycles of simulation. The lateral boundary conditions are created by combining and interpolating data from a non-tidal global model [2] and the TPXO tidal solution [17]. For every daily cycle, ReOMo checks if the current date coincides with the prescribed data assimilation date and carries out data assimilation when required. In the case of LD20_DA, LD60_DA, and LD120_DA, the data assimilation step is invoked every five days, although this frequency can be modified by the user. The SDD step of DA uses a large set of downscaling weights (in the case of LD20_DA it is about 125 million per variable), which are calculated off-line before the start of ReOMo. In the case of LD20_DA, it takes about 1.5 h to calculate the weights on a typical office PC.

The results of simulation are produced as 3-hourly instantaneous data, daily averages, and updated restart data for the next daily cycle. The output is archived using an SFTP transfer, and unnecessary intermediate files are removed by the housekeeping subsystem. A graphical representation of a typical daily cycle is shown in Figure 1.

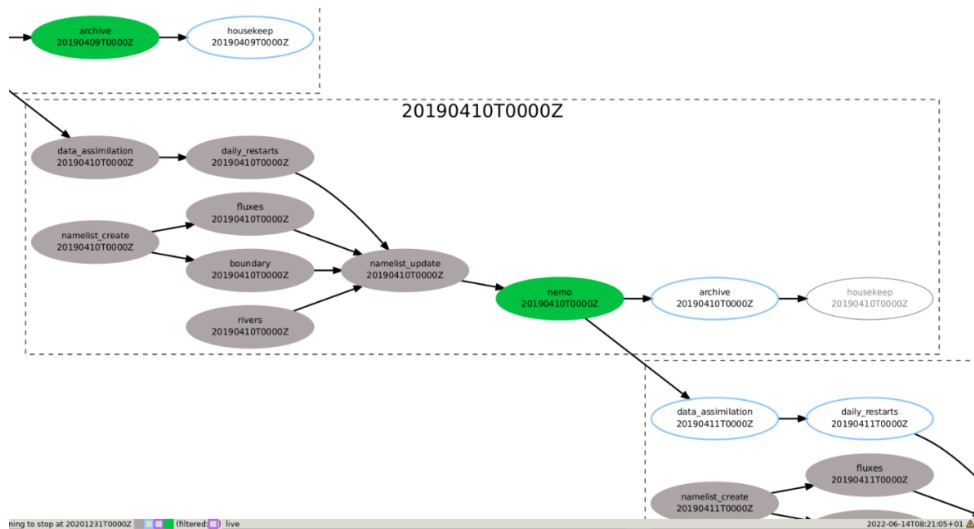

**Figure 1.** Graphical presentation of a daily cycle of the ReOMo workflow. The green colour highlights active tasks, completed tasks are shown in grey. The bubbles outlined blue are the tasks in waiting.

## 3. Results

### 3.1. The effect of Data Assimilation

Both versions of LD20 implementation of ReOMo were run for six years from 1 January 2015 to 31 December 2020. An important and innovative process used in ReOMo is the model-to-model data assimilation using the NDA algorithm. The steps of this process are illustrated in Figure 2 for 4 May 2016.

Table 1 shows differences introduced by model-to-model data assimilation for temperature and salinity at different depths. Data from three depth levels are presented: (i) d = 9.5 m representing the upper mixed layer; (ii) d = 53 m representing the seasonal thermocline; and (iii) d = 449 m representing the permanent thermocline. The data from CMEMS are vertically interpolated onto the LD20 depth levels, projected onto LD20 horizontal grid using SDD method and adjusted for time difference between daily average and midnight instantaneous readings.

In this example, the standard deviation of temperature between CMEMS and LD20_noDA is in the range 0.20–0.40 °C, while between CMEMS and LD20_DA it is much smaller, in the range of 0.04–0.11 °C. The reduction in standard deviations results from the reduction in bias achieved by the NDA method. The bias of LD20_noDA could be still acceptable for some applications. The main error in the free-run model is the spatial shift of physical features, which is discussed below.

The spatial distribution of Kalman gain coefficients for SST on the same day is shown in Figure 3.

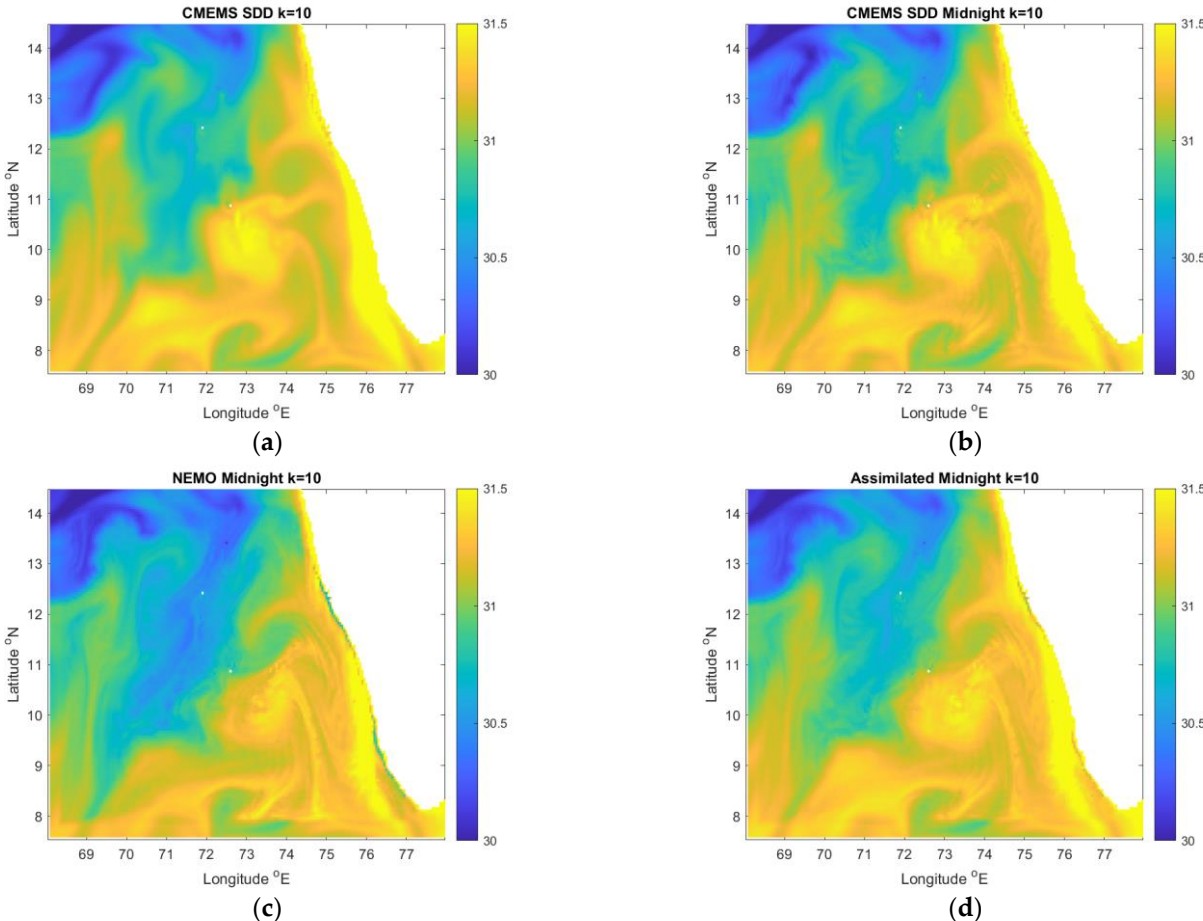

(**a**)
(**b**)
(**c**)
(**d**)

**Figure 2.** Temperature distribution at the depth 9.5 m on 4 May 2016: (**a**) daily average temperature from CMEMS model after linear interpolation vertically onto LD20 depth levels and downscaled onto LD20 horizontal grid using the SDD method; (**b**) same as (**a**) but after applying midnight correction to estimate temperature field at 24:00GMT on 4 May 2016; (**c**) instantaneous temperature at 24:00GMT on 4 May 2016 from LD20_DA before applying data assimilation at midnight on the same date; (**d**) the analysis state after assimilating data presented on the upper-right panel into LD20_DA at midnight.

**Table 1.** Standard deviation of temperature (°C) and salinity between various models.

| Variable | (Depth Level)/Depth | Model Output Pairs—All at 24:00GMT on 4 May 2016 | | | |
|---|---|---|---|---|---|
| | | CMEMS vs. LD20_DA (after DA Cycle) | CMEMS vs. LD20_noDA | LD20_noDA vs. LD20_DA (after DA Cycle) | LD20_DA before vs. LD20_DA after DA Cycle |
| Temperature | Surface | 0.07 | 0.25 | 0.23 | 0.09 |
| | (k = 10) 9.5 m | 0.08 | 0.27 | 0.24 | 0.08 |
| | (k = 20) 53 m | 0.11 | 0.40 | 0.38 | 0.09 |
| | (k = 25) 449 m | 0.04 | 0.20 | 0.20 | 0.03 |
| Salinity | Surface | 0.10 | 0.33 | 0.31 | 0.08 |
| | (k = 10) 9.5 m | 0.10 | 0.32 | 0.30 | 0.08 |
| | (k = 20) 53 m | 0.06 | 0.24 | 0.24 | 0.05 |
| | (k = 25) 449 m | 0.01 | 0.05 | 0.05 | 0.01 |

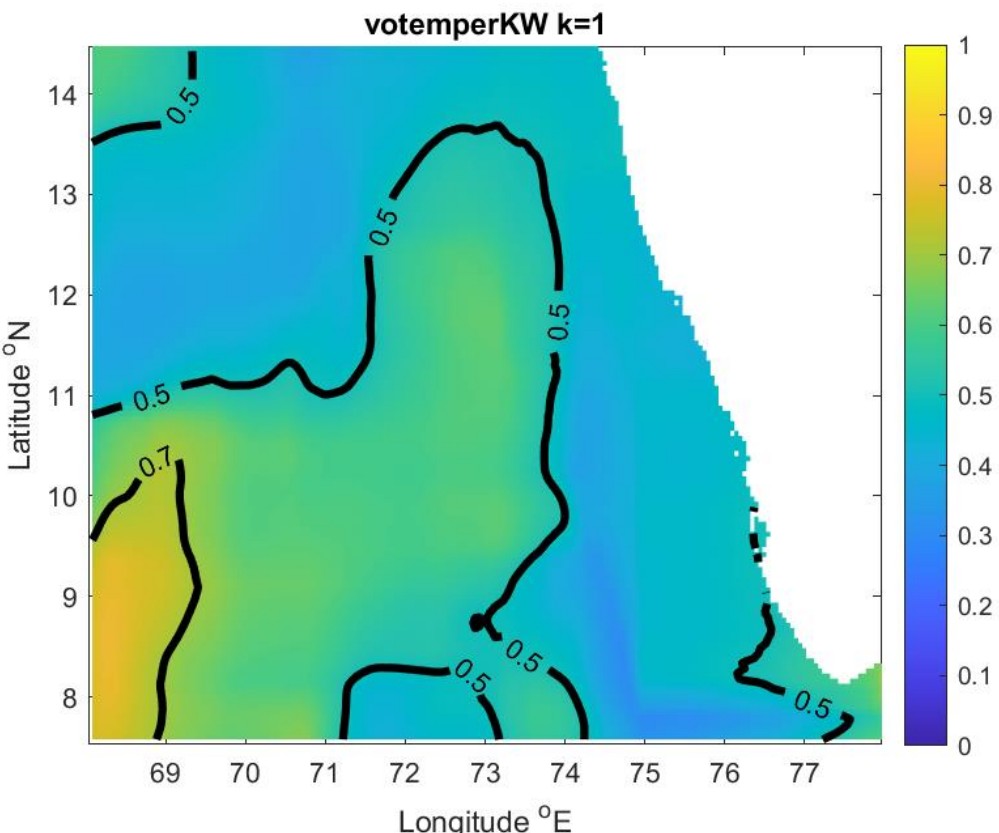

**Figure 3.** Spatial distribution of Kalman gain coefficient (*Kw*) for SST at midnight of 4 May 2016 as calculated using the procedure described in Section 2.3.

In the centre of the domain, the values of *Kw* calculated as described in Section 2.3 are close to 0.5, which indicates that the data from the external (CMEMS) and internal (LD20_DA) models have approximately equal contribution to the analysis state. In the southwest corner, the data from CMEMS have greater weight, and in the rest of the domain, the LD20 data prevail. The area-averaged Kalman gain weight *Kw* for SST in this example varies from *<Kw>* = 0.51 at the surface to *<Kw>* = 0.46 at 2235 m depth. Smaller-scale features resolved by the higher-resolution model (i.e., LD20_DA) propagate into the analysis state however with slightly reduced amplitude and provide a better granularity of model simulation. Larger-scale features are mainly taken from the coarser model. The area-average *<Kw>* for zonal and meridional current velocities at the surface are 0.38 and 0.42, respectively. A lower than 0.5 value of *Kw* and high correlation of the fields before and after DA cycle (in this case, Pierson correlation = 0.92) indicate that the DA cycle for currents predominantly takes finer-scale features from the higher-resolution model, and larger-scale features are taken from the coarser external model, similar to temperature and salinity.

Data assimilation helps reduce the spatial shift of physical features, which is also called 'the double penalty effect' [35]. This effect is common when comparing coarse and high-resolution models both in oceanography and meteorology. The ability of model-to-model data assimilation to reduce the phase shift is illustrated in Figure 4. Large-scale features that are resolved by the global 1/12° CMEMS model are presented in LD20_DA in a similar way. For example, the front between warmer and colder waters delineated by T = 31.3 °C isotherm is located at 72° E in the CMEMS model and at 72.25° E in LD20_DA. In contrast, the LD20_noDA model shows the front shifted to the west by nearly 150 km to 70.5° E.

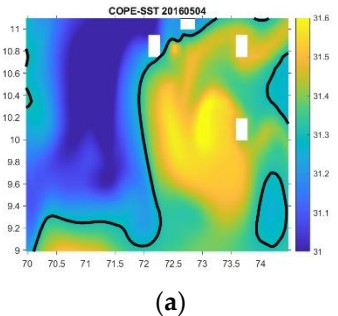　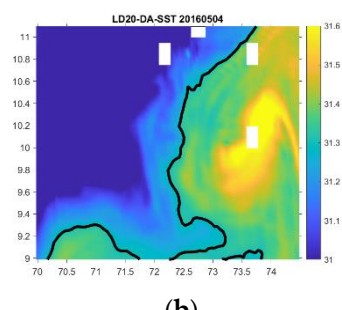　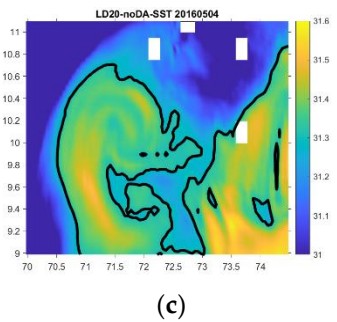

(**a**)　　　　　　　　　　　　　　(**b**)　　　　　　　　　　　　　　(**c**)

**Figure 4.** Sea surface temperature for the three models in a zoomed area south of Lakshadweep islands on 4 May 2016: (**a**) CMEMS, (**b**) LD20_DA with 5 daily cycle of data assimilation, (**c**) LD20_noDA without data assimilation. Black contour shows the 31.3 °C isoline.

The computational efficiency of ReOMo can be seen from the following timings obtained from the Rose-Cylc statistical utility. A typical time for NEMO LD20 computation (both DA and noDA versions) on a mesh 200 × 140 × 50 for one day forecast is 2 min, the data assimilation cycle (every five days) is 2.5 min, and preparation of boundary conditions, which is done in advance to save time, is about 2 min. Other jobs such as creating and updating the namelist, copying daily restarts, archiving the results onto external storage, and cleaning the working space on the cluster, take 5–15 s each. The full 5-day ReOMo forecasting cycle for LD20_DA including data assimilation lasts 14 min on 71 computing cores. For LD60_DA (on a mesh 600 × 420 × 50), the full 5-day cycle takes 100 min including 16 min for data assimilation. For LD120_DA (over a smaller domain on a mesh 300 × 300 × 50) the full 5-day cycle takes 85 min including 4 min for data assimilation. LD120 models were run on 100 computing cores and used the DA code that was optimised for speed.

*3.2. Model Validation*

The LD20 implementation of ReOMo was validated against OSTIA (SST), Argo floats (temperature and salinity profiles), and data from the local weather buoys (SST). It was also verified against data assimilating Copernicus global model GLOBAL_REANALYSIS_PHY_ 001_030-TDS. Comparisons are presented for domain averaged magnitudes for the models, OSTIA and GHR-MUR, and for point magnitudes at the location of the buoys and Argo floats. Figure 5 shows a time series of area-averaged SST from LD20_noDA, LD20_DA, CMEMS, OSTIA, and GHR-MUR. For comparison, the data from a non-assimilating version of LD20_noDA model is also shown. As can be seen, the data assimilation reduces differences between the model and observations.

The improvement provided by model-to-model data assimilation is particularly seen during the monsoon period (July–September) 2015. The solar radiation used to drive the LD20 models was overestimated during this period and corrected for later dates by the meteorological data supplier [36]. In response to overestimated heat flux, the LD20_noDA was significantly higher, up to 1.5 °C SST compared to OSTIA, while the data assimilating model LD20_DA only shows a difference of up to 0.5 °C.

The uncertainty analysis includes the estimates of bias, root-mean-square differences, and point-to-point correlations. Table 2 shows model bias, RMSD, and Pierson correlation coefficient, which are calculated using daily averaged SST data as follows:

$$Bias(t) = \frac{1}{N} \sum_{i=1}^{N} (M_i(t) - O_i(t))$$

$$RMSD(t) = \left[ \frac{1}{N} \sum_{i=1}^{N} (M_i(t) - O_i(t))^2 \right]^{1/2}$$

$$Corr(t) = \frac{1}{N} \sum_{i=1}^{N} \frac{(M_i(t) - M(t))(O_i(t) - O(t))}{\sigma_M(t)\sigma_O(t)}$$

where $M_i(t)$ and $O_i(t)$ are model and observational data at grid node number $i$, $M(t)$, $O(t)$, $\sigma_M(t)$, and $\sigma_O(t)$ are their area averages and standard deviations, $t$ is time, and $N$ is the total number of wet grid nodes at the sea surface. Then, these statistics were additionally averaged over six years of simulations from 2015 to 2020. Calculations excluded a narrow rim of grid points near the boundary where the flow relaxation scheme was applied. All statistics were calculated taking OSTIA as a basis. For comparison, the same statistics were calculated for an alternative observational data set GHR-MUR, as well as for the LD20_noDA model, which was run without data assimilation.

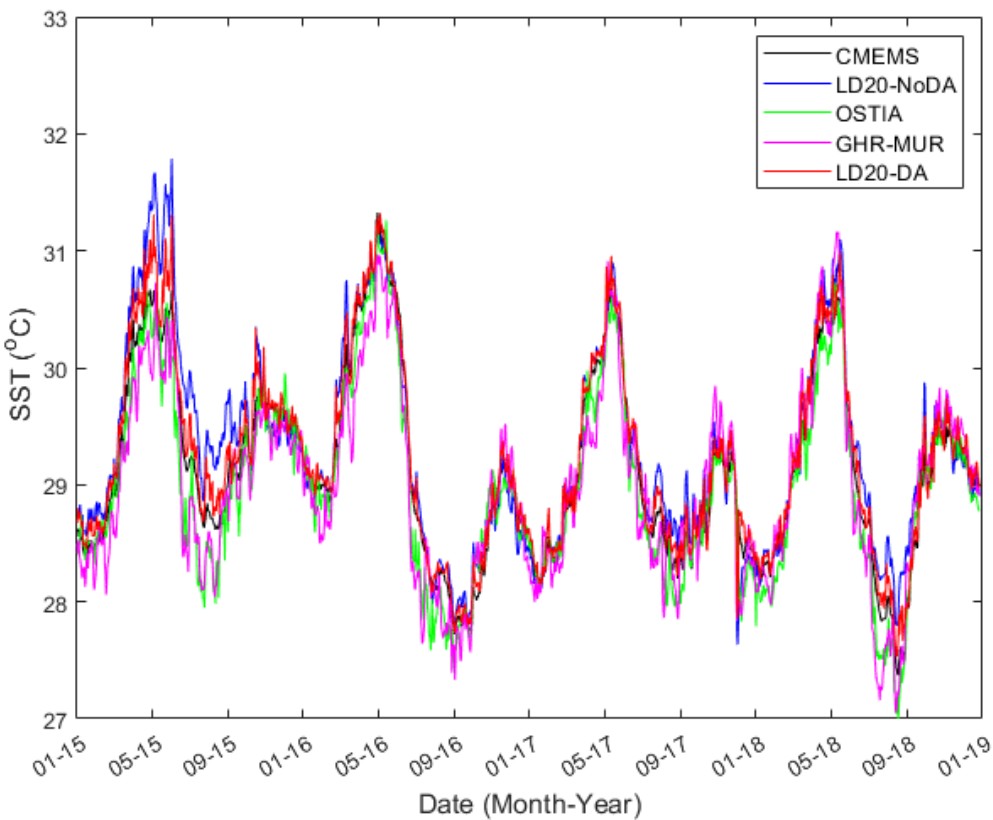

**Figure 5.** Time series of area-averaged SST from LD20_noDA, LD20_DA in comparison to OSTIA and GHR-MUR observations, as well as CMEMS global reanalysis.

**Table 2.** Statistics showing the skills of models in representing SST.

| Ref. Data OSTIA | | LD20_DA | CMEMS | GHR-MUR | LD20_noDA |
|---|---|---|---|---|---|
| Average over year 2015–2018 | RMSD, °C | 0.42 | 0.35 | 0.38 | 0.53 |
| | BIAS, °C | 0.23 | 0.14 | 0.01 | 0.31 |
| | Corr. | 0.61 | 0.65 | 0.67 | 0.53 |

OSTIA and LD20 have the same spatial resolution (1/20°); therefore, the RMSD is a good indicator of model skill to represent smaller scale features not resolved by the coarser external model. In terms of RMSD, the best accuracy is achieved by the CMEMS, with LD20_DA being not far away. When comparing the data sets, it must be taken into account that CMEMS, OSTIA, and GHR-MUR share the same data sources; therefore, it is expected that the difference between them should be small. LD20_DA clearly shows an improvement compared to LD20_noDA in all skill parameters. It gives results closer to OSTIA even though OSTIA is not directly assimilated into LD20_DA, only via the global model CMEMS. This is a result for which we aimed when applying mode-to-model data assimilation. The deviations between LD20_DA and OSTIA are within the accuracy of

the OSTIA data itself (0.57°, see [37]), and is also similar to the deviations between two alternative observational data sets, OSTIA and GHR-MUR.

The spatial distribution of the models' skill was assessed by comparing sea surface temperature (SST) produced by models with the OSTIA data set, which produces high-resolution analysis for the SST of the global ocean from satellite and in situ data. The maps of models' skill (see Figure 6) were computed by calculating bias and root-mean-square difference at each grid point cell taking OSTIA as the basis. Data sets on different horizontal meshes were interpolated onto the LD20 grid.

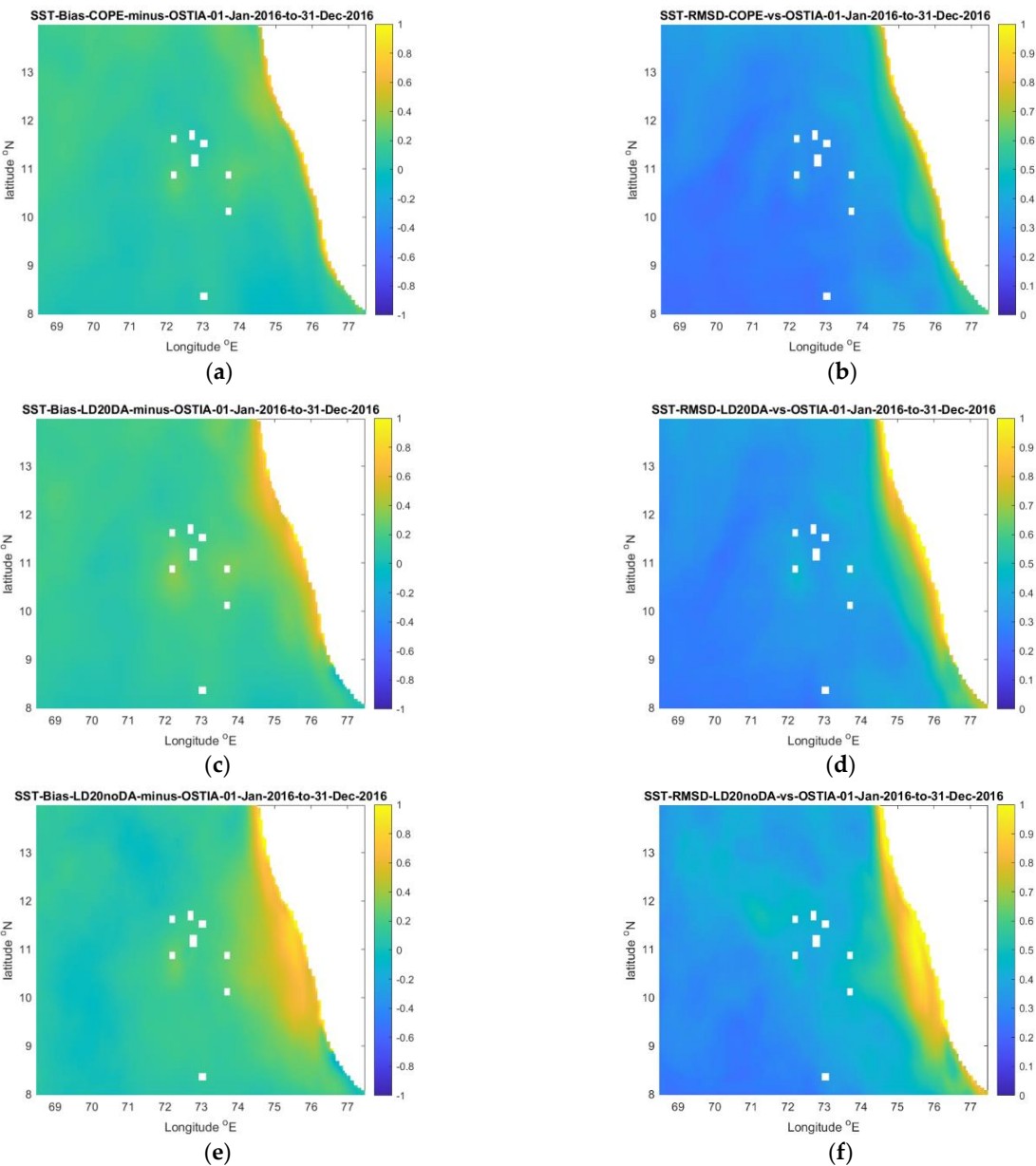

**Figure 6.** Spatial distribution of time-averaged differences between the models and OSTIA: (**a,c,e**) bias; (**b,d,f**) RMSD; (**a,b**) CMEMS; (**c,d**) LD20_DA, (**e,f**) LD20_noDA. Time-averaging period is from 01 January to 31 December 2016.

The model skill of LD20_DA was further assessed by computing the spatial distribution of the Willmott skill parameter; see, e.g., [38,39], defined by the equation:

$$WS_i = 1 - \frac{\langle (M_i - O_i)^2 \rangle}{\langle (|M_i - \langle O_i \rangle| + |O_i - \langle O_i \rangle|)^2 \rangle}$$

where angle brackets denote time averaging and vertical bars denote absolute values. The Willmott skill parameter is a simple measure of the agreement between two data sets; WS = 1 indicates a perfect match, whereas WS = 0 means there is no agreement at all. As an example, the maps of WS averaged over 2016 are presented in Figure 7 for LD20_DA and CMEMS. The WS for LD20_DA is very close to one (and to the WS for CMEMS) everywhere in the domain, showing that assimilation of observational SST via the global model works as expected.

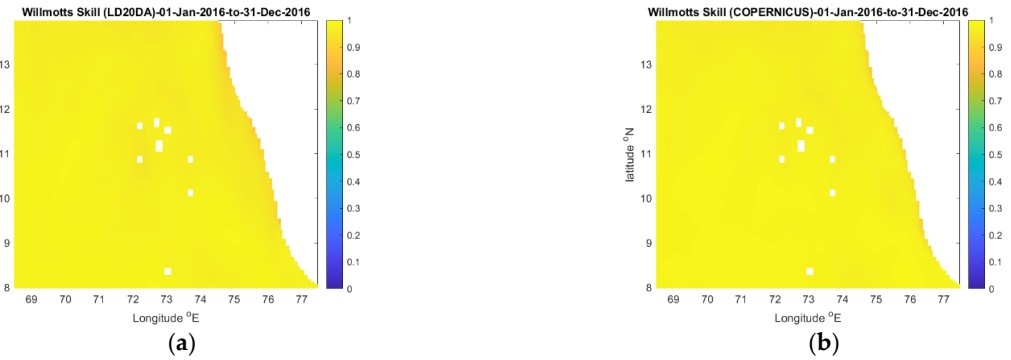

(**a**)  (**b**)

**Figure 7.** Spatial distribution of Willmott skill parameter averaged for year 2016 with OSTIA data taken as reference: (**a**) LD20_DA; (**b**) CMEMS.

The accuracy of ReOMo within the water column was assessed by comparing temperature and salinity profiles with observed values available from Argo floats [20]. For this analysis, a total number of 401 Argo profiles was used covering a period from 2015 to 2020; see Figure 8.

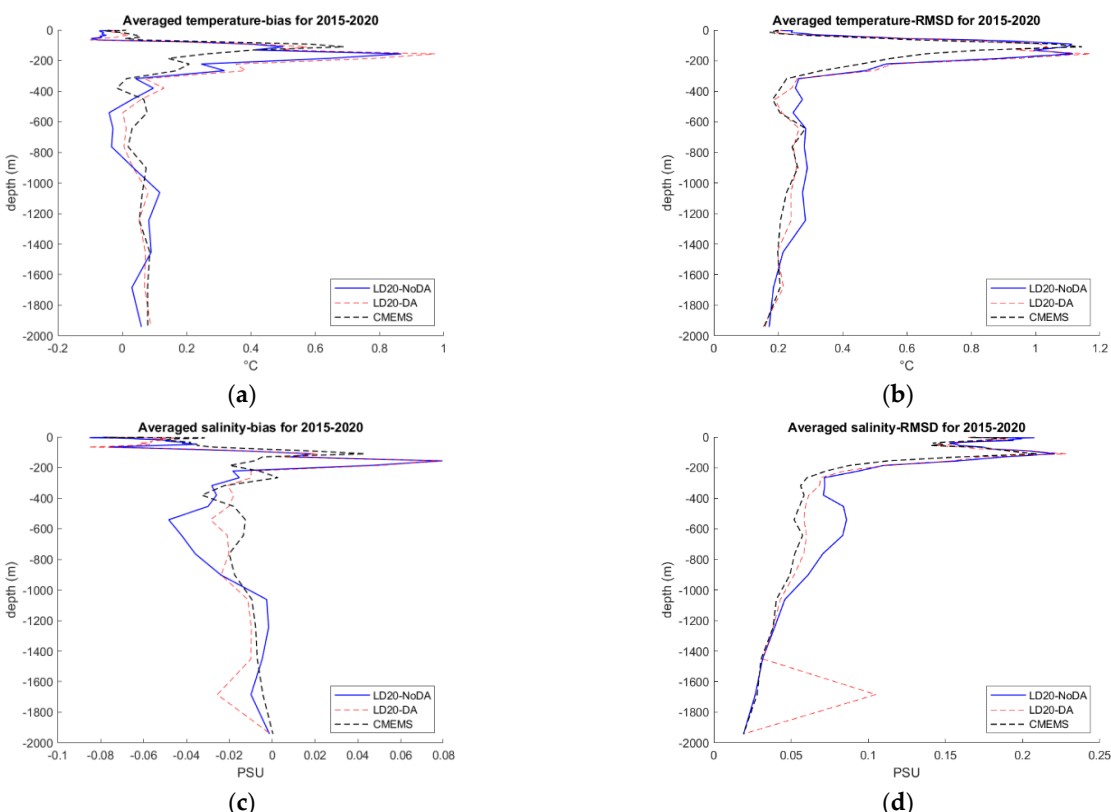

(**a**)  (**b**)

(**c**)  (**d**)

**Figure 8.** Comparison of LD20_DA, LD20_noDA and CMEMS against Argo float profiles of temperature and salinity: (**a**,**c**) bias; (**b**,**d**) RMSD; (**a**,**b**) temperature; (**c**,**d**) salinity. Potential temperature produced by the models is converted to in situ values measured by floats using the Thermodynamic Equation of Seawater-2010. Averaging is done over 401 individual Argo profiles for the period 1 January 2015 to 31 December 2020.

The uncertainty generated by LD20_DA is similar to that produced by CMEMS reanalysis. For both models, the largest uncertainty is within the depth range of 150–250 m. At lower depth levels, the RMSDs reduce to 0.2–0.3 °C for temperature and 0.05–0.1 PSU for salinity. LD20_DA is likely to inherit positive deviations in both temperature and salinity from CMEMS data used for boundary conditions. This view is supported by a similar behaviour of LD20_no DA, which only accepts boundary conditions from CMEMS but does not assimilate its 3D data in the interior.

Table 3 shows how SST produced by LD20_DA compares with observational data obtained from three moored weather buoys located within 1 to 5 miles of the islands of the Lakshadweep archipelago. The data were obtained from [40]. The RMSD and bias were calculated using the entire available period of observation.

**Table 3.** Comparison of SST from LD20_DA and moored weather buoys.

| Buoy ID | Latitude, °N | Longitude, °E | Period | Bias, °C | RMSD, °C |
|---|---|---|---|---|---|
| MB2300454 | 10.32 | 72.59 | 27 October 2016 to 31 December 2020 | −0.08 | 0.29 |
| MB2300492 | 10.87 | 72.21 | 23 May 2016 to 31 December 2020 | 0.22 | 0.44 |
| MB2300497 | 10.61 | 72.30 | 23 May 2016 to 31 December 2020 | 0.07 | 0.44 |

In all cases, the RMSD is less than 0.45 °C, which is slightly better than the uncertainty of RMSD = 0.57 °C provided by OSTIA. The seasonal variation of SST from LD20_DA in comparison with buoy measurements is illustrated in Figure 9 for year 2018.

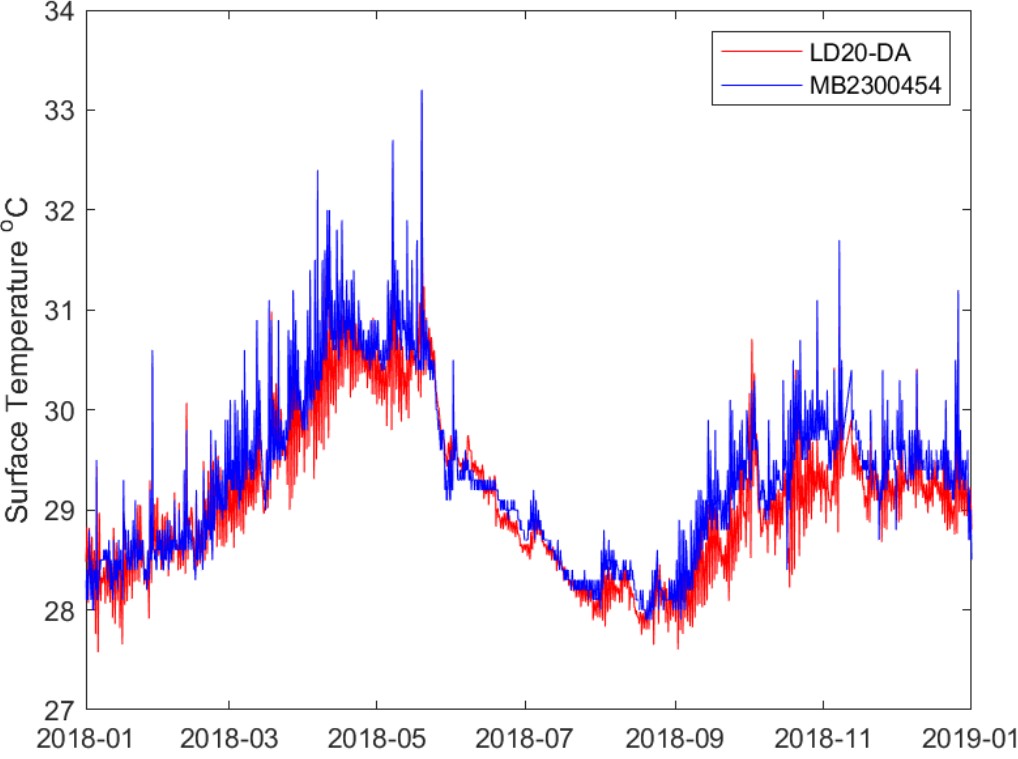

**Figure 9.** Time series of SST from LD20_DA and weather buoy MB2300454 located south of Kavaratti Island at 10.32° N, 72.59° E from 1 January 2015 to 31 December 2015. The model SST is bi-linearly interpolated to the location of the buoy.

### 3.3. Higher Resolution Models

The higher resolution models (LD60 and LD120) provide better granularity, as expected. As with many other high-resolution models without data assimilation, they are prone to the spatial phase shift. The shift is significantly reduced by model-to-model DA using the ReOMo NDA algorithm. Figure 10 shows the current velocity distribution from the CMEMS global model, LD60_noDA, and LD60_DA.

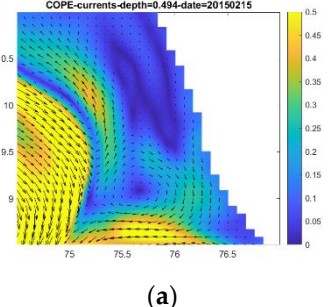
(**a**)

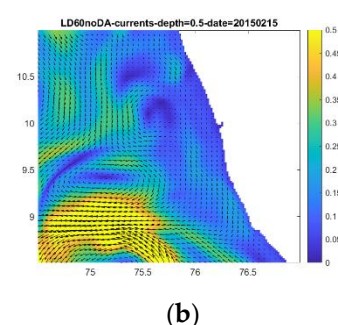
(**b**)

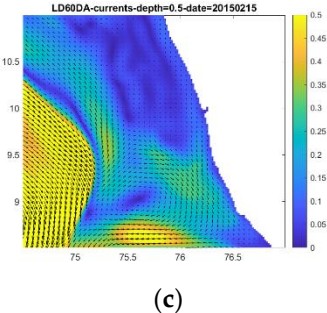
(**c**)

**Figure 10.** Daily averaged current velocities at the sea surface on 2 February 2015: (**a**) CMEMS model at 1/12° resolution; (**b**) LD60_noDA at 1/60°; (**c**) LD60_DA also at 1/60°. Colour shows the speed of current in m/s.

The anticyclonic eddy centred at 9.1° N, 75.7° E is well resolved by the global data assimilating model. This eddy is displaced to 9.4° N, 75.2° E in LD60_noDA simulations. It is restored to its correct position in the data assimilating model LD60_DA. A similar effect is seen for the LD120 (1/120° resolution) models. The anticyclone is shifted to 9.3° N, 75.8° E in the model without DA and is restored to its correct location in the data assimilating LD120_DA model; see Figure 11.

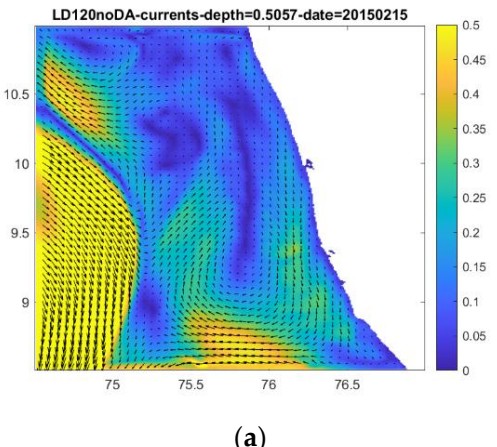
(**a**)

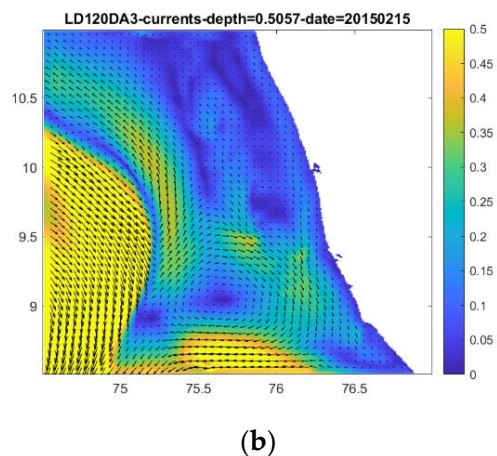
(**b**)

**Figure 11.** Daily averaged current velocities at the sea surface on 2 February 2015: (**a**) LD120_noDA; (**b**) LD120_DA. Colour shows the speed of current in m/s.

### 4. Discussion

Dynamical downscaling is generally considered as the approach consisting of running a high-resolution regional simulation ('child' models) inside a subdomain covered by a 'parent' lower-resolution model, see e.g., [6]. The benefits of downscaled regional models have been discussed in a number of studies; see e.g., [4,41] and references therein. A practical need of having a 'rapid-response' high-resolution model was formulated and discussed by Onken et al. [42]. However, setting up a data assimilating regional model in a new domain, particularly in operational mode, takes time. Therefore, some high-resolution nested models do not use data assimilation at all; see e.g., [43]. Non-assimilating models are suitable for process studies but are usually not accurate enough for operational use. Aguiar et al. [6] concluded that, in the absence of data assimilation, their downscaled model

was 'not able to correctly locate the mesoscale structures, which negatively affects the representation of the variability of surface currents'. This paper presents a rapid response 'Crisis Ocean Forecasting System' which is quick to set up, assimilates data from the freely available coarse model reanalysis and forecast products, and automates many routine and intermittent tasks.

This study uses the NEMO modelling engine, which has been successfully and widely exploited both for research and operational purposes. A critical component of a modern regional ocean model is an efficient data assimilation component, which helps keep the model in agreement with reality. Existing data assimilation procedures are based on a number of assumptions and simplifications (see e.g., [21,23]), and only practical applications can judge if these assumptions are acceptable. The innovative feature of this study is the use of a model-to-model data assimilation approach in an operational environment. Therefore, the focus here is on the efficiency of data assimilation.

It is known that spurious high-frequency oscillations occur in forecasts made with the primitive equations if the initial fields of density and velocity are not in an appropriate state of balance with each other [44]. Density and velocity data gained from unrelated observations are not in a dynamic balance and an additional effort is required to filter out high-frequency oscillations [45]. Current velocity data produced by a global model are dynamically balanced with temperature and salinity fields by design. Therefore, the assimilation of 3D rather than 2D model data has an additional benefit of removing the need for dynamic balancing or filtering data coming directly from observations.

The relocatable operational forecasting system titled ReOMo was designed, implemented, and tested for the case of Lakshadweep Sea in a domain of approximately $1100 \times 700$ km$^2$. This paper also briefly describes higher-resolution models, which were run within ReOMo at $1/60°$ and $1/120°$ resolutions. Two versions of the LD20 model were developed, LD20_DA and LD20_noDA—with and without data assimilation respectively. Both versions were run for an extended period, from 1 January 2015 to 31 December 2020, in order obtain reliable statistics of model performance. Models were validated against various observational data sets—the Operational Sea Surface and Ice Analysis, Argo float profiles, and moored weather buoys. The LD20_DA performed consistently better than LD20_noDA, confirming that the new DA system helps improve the model outputs.

Area-average statistics, which are more dependent on model accuracy than resolution, show close performance of LD20_DA and the state-of-the-art EU CMEMS reanalysis. Both CMEMS and LD20_DA demonstrate better agreement with OSTIA in the winter months than in June–September. This is likely caused by degradation of accuracy of satellite-derived data due to heavy clouds and rain during the south-west monsoon period. Comparison with Argo float profiles shows good agreement except at the depth range of 150–250 m. This discrepancy in this depth range is seen in both LD20 model and CMEMS reanalysis. LD20_DA is likely to inherit positive deviations in temperature and salinity from CMEMS data used for boundary conditions. This view is supported by a similar behaviour of LD20_noDA, which only accepts boundary conditions from CMEMS but does not assimilate its 3D data in the interior. ReOMo is computationally efficient. In our experiments, the five-day analysis-forecasting cycle took between 14 and 100 min depending on the number of computing nodes, domain size, and model resolution: $1/20°$, $1/60°$, or $1/120°$. Our tests show that ReOMo can be used as a Crisis Ocean Modelling System due to its ease of implementation and good accuracy at medium and high resolution.

## 5. Conclusions

This paper presents the Relocatable Operational Ocean Model (ReOMo), which can be quickly nested into a larger area model and used as a Crisis Ocean Modelling System in any region of the global ocean that is free from ice. The core components of ReOMo are the NEMO hydrodynamic model and Rose-Cylc workflow management software. A user is required to prepare initial data only once, after which the system works automatically. The principal innovative feature of ReOMo is the use of Nesting with Data Assimilation (NDA),

which is based on the model-to-model assimilation technique. The NDA utilises the full 3D set of field variables (temperature, salinity, and zonal and meridional current velocities) from the external, e.g., the global model rather than just the 2D boundary conditions. Therefore, ReOMo becomes physically aware of observations that are assimilated and dynamically balanced in the external model. The NDA also reduces the spatial phase shift of ocean features, also known as the 'double penalty effect'. In this study ReOMo was implemented for the Lakshadweep Sea in the Indian Ocean at $1/20°$, $1/60°$, and $1/120°$ resolution, with and without model-to-model data assimilation. ReOMo is computationally efficient; one data assimilation cycle takes approximately the same time as one day of free-running the NEMO model. ReOMo was validated against OSTIA, Argo floats, and moored weather buoy data, which showed it has accuracy similar to that of the state-of-the-art CMEMS global model with an additional benefit of having better resolution.

**Author Contributions:** Conceptualization, G.I.S.; Formal analysis, G.I.S. and J.M.G.-O.; Funding acquisition, G.I.S.; Methodology, G.I.S. and J.M.G.-O.; Resources, G.I.S.; Software, G.I.S., J.M.G.-O., M.S., J.T. and M.A.; Supervision, G.I.S.; Validation, G.I.S. and M.S.; Visualization, G.I.S. and M.S.; Writing—original draft, G.I.S.; Writing—review and editing, G.I.S. and J.M.G.-O. All authors have read and agreed to the published version of the manuscript.

**Funding:** This research was funded by the University of Plymouth Enterprise LTD.

**Institutional Review Board Statement:** Not applicable.

**Informed Consent Statement:** Not applicable.

**Data Availability Statement:** External data are available from their respective providers. Data generated by the authors are available on reasonable request.

**Conflicts of Interest:** The authors declare no conflict of interest.

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
