# Peer review of "Crisis Ocean Modelling with a Relocatable Operational Forecasting System and Its Application to the Lakshadweep Sea (Indian Ocean)"

_jmse, doi:10.3390/jmse10111579_

Round 1
Reviewer 1 Report
This manuscript describes a process for local fine-resolution model simulation of oceanic fields, guided by boundary conditions from a coarser-resolution wider-area model and assimilation (in the local model interior) of the wider model fields (which may themselves be guided by / assimilate observed data). Much “machinery”, largely based on existing published processes, is involved in the assimilation, the sequencing of model steps and introducing observed and contextual data to set up and run the system to simulate real conditions in time and space. There is extensive description of validation and comparison between results with and without the assimilation in the local model interior – assimilation improves the results. I think this should be of interest to organisations such as CMEMS with an interest in operational downscaling – to resolution O(1 km) – from wider-area models with coarser resolution. I do not know how far they might have developed their own processes for doing this. The manuscript also claims that the process is suitable for rapid application but I suppose the speed of implementation will depend very much on the experience, skill and practice of the practitioner.
Inevitably for a readable manuscript, much of the detail about the “machinery” is left to many references. The scientific criterion of being able to repeat this work, and the practical criterion of being able to implement the process with all its “machinery”, would only be feasible for organisations already well-versed in the literature and practised in the field (model runs with some form of data assimilation and use of observed and contextual data to simulate real conditions in time and space).
Broadly, I think this manuscript is publishable with some improvements. The “Specific comments” below are mostly meant to clarify intended meaning. Additionally, some copy editing could improve the use of English. Two of the specific comments (“Line 167”, “Lines 336-349”) are more substantive in calling for additional discussion of what I believe to be significant questions. There is also a journal policy question in “Line 631”.
Specific comments.
Abstract
Line 21. Better “. . aware of any observations which may be assimilated . .” Present wording suggests that the parent model assimilates observations which I suppose might not be the case.
Introduction
Lines 36-37. Maybe “DIgital Twins of The Ocean (DITTO)”
Line 60. Introduce acronym CAMs here.
Line 63. “the set up” –> “setting up”
Section 2.
Line 112. “usual Flather radiation conditions” – some readers may want a reference to find these.
Line 132. “. . UK Met Office”
Lines 140-141. “which is a data assimilating model on its own and” –> “which is itself a data assimilating model and” or “which is a data assimilating model in its own right and”
Line 167. What is the basis of choosing 5 days for the DA cycle?
Section 3
Lines 336-349. The KW values (balance between external and internal models’ contributions to the analysis state) seem to be “accepted”. I think there should be some discussion about what the balance should be and how the process might achieve the desired balance. [If the KW values are somehow the result of the process achieving an optimal balance, then this should be explained.]
Lines 376-377. Better “. . and used DA code optimised for speed.”?
Lines 381, 383, 389, figure 5, Table 2 and later. GHR-MUR: do you mean GHRSST-MUR as in lines 133, 134, 412 and later? Otherwise I think GHR-MUR needs definition.
Lines 436-437. “. . December 2016.” [But this sentence may be unnecessary as it repeats the figure caption.]
Line 453 “averaged over 2015-2018” contradicts figure 7 caption line 460 “averaged for year 2016”
Lines 521-522. Better “As” –> “Like” or “As with” and move “without data assimilation” to after “models”.
Line 523. “by” –> “the”.
Line 537. “9.2” –> “9.4”?
Line 631 “Data Availability Statement: Not applicable.” I leave it to JMSE whether this is acceptable. Much external data are used in setting up and running the model; the issue may be whether model output (especially as used in figures and tables) is regarded as data that should be available.
Author Response
Comment:
This manuscript describes a process for local fine-resolution model simulation of oceanic fields, guided by boundary conditions from a coarser-resolution wider-area model and assimilation (in the local model interior) of the wider model fields (which may themselves be guided by / assimilate observed data). Much “machinery”, largely based on existing published processes, is involved in the assimilation, the sequencing of model steps and introducing observed and contextual data to set up and run the system to simulate real conditions in time and space. There is extensive description of validation and comparison between results with and without the assimilation in the local model interior – assimilation improves the results. I think this should be of interest to organisations such as CMEMS with an interest in operational downscaling – to resolution O(1 km) – from wider-area models with coarser resolution. I do not know how far they might have developed their own processes for doing this. The manuscript also claims that the process is suitable for rapid application but I suppose the speed of implementation will depend very much on the experience, skill and practice of the practitioner.
Inevitably for a readable manuscript, much of the detail about the “machinery” is left to many references. The scientific criterion of being able to repeat this work, and the practical criterion of being able to implement the process with all its “machinery”, would only be feasible for organisations already well-versed in the literature and practised in the field (model runs with some form of data assimilation and use of observed and contextual data to simulate real conditions in time and space).
Broadly, I think this manuscript is publishable with some improvements. The “Specific comments” below are mostly meant to clarify intended meaning. Additionally, some copy editing could improve the use of English. Two of the specific comments (“Line 167”, “Lines 336-349”) are more substantive in calling for additional discussion of what I believe to be significant questions. There is also a journal policy question in “Line 631”.
Response:
Thanks.
It is true that “the speed of implementation will depend very much on the experience, skill and practice of the practitioner”. However this does not contradict our claim of being a “rapid-response” model. Our method is simpler and requires much less pre-processing effort, and hence, being other considerations identical (type of problem, experience and knowledge of the practitioner, etc.), it is faster to implement than standard, data assimilated models.
We address the specific comments, one by one below.
Specific comments.
Abstract
Comment: Line 21. Better “. . aware of any observations which may be assimilated . .” Present wording suggests that the parent model assimilates observations which I suppose might not be the case.
Response: The parent does assimilate observations. We made it more clear by rewording as follows: ‘Therefore ReOMo becomes physically aware of observations which have been assimilated and dynamically balanced in the external model’.
Introduction
Comment: Lines 36-37. Maybe “DIgital Twins of The Ocean (DITTO)”
Response: Amended as advised.
Comment: Line 60. Introduce acronym CAMs here.
Response: Amended as advised.
Comment: Line 63. “the set up” –> “setting up”
Response: We do not use the wording ‘the set up’. Instead we use ‘ReOMo can be quickly set up in a new area…’
Section 2.
Comment: Line 112. “usual Flather radiation conditions” – some readers may want a reference to find these.
Response: As advised, the reference to the NEMO user guide is added which contains a detailed description of Flather radiation conditions.
Comment: Line 132. “. . UK Met Office”
Response: As advised, the word ‘UK’ has been added.
Comment: Lines 140-141. “which is a data assimilating model on its own and” –> “which is itself a data assimilating model and” or “which is a data assimilating model in its own right and”
Response: Amended as advised.
Comment: Line 167. What is the basis of choosing 5 days for the DA cycle?
Response: This is the outcome of a few sensitivity tests and is a compromise between an improved accuracy (if DA is carried out frequently, say every day) and computational efficiency (when DA is carried out infrequently). Our models run free (without DA) with a very small error for at least 10 days. The 5-day cycle was selected to be on a safe side. Clarification is given in new lines 168-169.
Section 3
Comment: Lines 336-349. The KW values (balance between external and internal models’ contributions to the analysis state) seem to be “accepted”. I think there should be some discussion about what the balance should be and how the process might achieve the desired balance. [If the KW values are somehow the result of the process achieving an optimal balance, then this should be explained.]
Response: Yes, the KW (renamed now as Kw for consistency with Section 2.3) values are the result of a usual process (see e.g. [21]) of calculating Kalman gain coefficients based on the variances of fluctuations produced by the internal and external model. This process is described in lines 224-230 of the original MS, and in more detail in reference [8]. The optimal balance is achieved as this procedure minimises the cost function Js as described in lines 206-224 of the original MS. The Figure 3 captions and the text between the lines 340 and 354 (revised MS) is amended to clarify this as advised, and a reference to Section 2.3 is given where the actual calculation procedure is presented.
Comment: Lines 376-377. Better “. . and used DA code optimised for speed.”?
Response: Amended as advised.
Comment: Lines 381, 383, 389, figure 5, Table 2 and later. GHR-MUR: do you mean GHRSST-MUR as in lines 133, 134, 412 and later? Otherwise I think GHR-MUR needs definition.
Response: Thank you. The name of the data set has now been standardised as GHR-MUR throughout the MS.
Comment: Lines 436-437. “. . December 2016.” [But this sentence may be unnecessary as it repeats the figure caption.]
Response: The repetition removed as advised
Comment: Line 453 “averaged over 2015-2018” contradicts figure 7 caption line 460 “averaged for year 2016”
Response: Thank you. Corrected as advised. The correct averaging period is 01 Jan to 31 Dec 2016.
Comment: Lines 521-522. Better “As” –> “Like” or “As with” and move “without data assimilation” to after “models”.
Response: Amended as advised.
Comment: Line 523. “by” –> “the”.
Response: Amended as advised.
Comment: Line 537. “9.2” –> “9.4”?
Response: Thank you. Of course it should be 9.4N. Amended as advised.
Comment: Line 631 “Data Availability Statement: Not applicable.” I leave it to JMSE whether this is acceptable. Much external data are used in setting up and running the model; the issue may be whether model output (especially as used in figures and tables) is regarded as data that should be available.
Response: Amended to read: ‘External data are available from their respective providers. Data generated by the authors are available on reasonable request’.
Reviewer 2 Report
I thank the authors and moving to the paper itself. As far as authors are presenting:
ReOMo is used as a Crisis Ocean Modelling System in any region of the World Ocean
I am sorry but I have a number of problems with this paper, broadly:
1. Please change the title to “Crisis ocean modelling with a relocatable operational forecasting system in Lakshadweep Sea: A case study”
2. You are applied the existing model. You should compared the result with existing applied model
3. NDA algorithm is prepared by you or you used the universal one. If it is developed by you kindly add it in manuscript or supplementary file.
4. You have taken three different types of depth level (9.5, 53, 449m) in Table 1. Why three not five or six. Why particularly these three.
5. In model validation author used only temp., why not for salinity.
6. Author must add uncertainty analysis in validation section. As this model is an existing model.
7. In Table 3, bias value is in –ve; is there any specific reason for that.
Author Response
Reviewer 2: I thank the authors and moving to the paper itself. As far as authors are presenting:
ReOMo is used as a Crisis Ocean Modelling System in any region of the World Ocean
I am sorry but I have a number of problems with this paper, broadly:
Comment: 1. Please change the title to “Crisis ocean modelling with a relocatable operational forecasting system in Lakshadweep Sea: A case study”
Response: The MS represents two components of our study: (ii) the development of a relocatable operational forecasting system and (ii) application of this system to the Lakshadweep Sea, therefore the word ‘and’ is very important. The words ‘Indian Ocean‘ are also important as the Lakshadweep Sea is a less know domain outside the Indian oceanographic community. Our choice of words: “its application to the Lakshadweep Sea” already suggest that the article contains a case study, but not just that. We believe that the existing title is correctly represents the content of the MS.
Comment: 2. You are applied the existing model. You should compared the result with existing applied model
Response: We have developed a set of new models LD20_noDA, LD20_DA, LD60_DA, LD120_DA which are based on the NEMO v3.6 codebase. The models are compared with observations and with the existing model (also based on NEMO code) , namely GLOBAL_REANALYSIS_PHY_001_030-TDS, see Figure 2 , Table 1 and following sections of the MS. The text in lines 377-389 is amended to clarify this point.
Comment: 3. NDA algorithm is prepared by you or you used the universal one. If it is developed by you kindly add it in manuscript or supplementary file.
Response: The NDA algorithm has been developed in the study presented in reference [8] which is cited on pages 65, 141, 156, 235. In order to avoid repetition, we refer to these results rather than reproduce them in full in the current MS. We amended the text in line 65 to clarify this point.
Comment: 4.You have taken three different types of depth level (9.5, 53, 449m) in Table 1. Why three not five or six. Why particularly these three.
Response: The following clarification is added to the MS (new lines 314-316) as advised. ‘Data from three depth levels are presented: (i) d=9.5 m representing the upper mixed layer, (ii) d=53 m representing the seasonal thermocline, and (iii) d= 449m representing the permanent thermocline. ‘
Comment: 5. In model validation author used only temp., why not for salinity.
Response: Validation is carried out with temperature and salinity wherever possible. For example, OSTIA, Mbuoys and GHR-MUR data sets do not contain salinity, while ARGO float do. Comparison with Argo float salinity profiles is presented in Figure 8(c) and Figure 8(d). The text in lines 386-387 is modified to clarify this point.
Comment: 6. Author must add uncertainty analysis in validation section. As this model is an existing model.
Response: The uncertainty analysis is presented in the form of standard deviations in Table 1, RMSD and Bias in Table 2, RMSD and Bias in Table 3, Figure 6 (a-d) , Willmott Skill parameter in Figure 7(a) and Figure 7(b), Bias and RMSD in Figure 8 (a-d). Clarification is given in lines 407-408 of the revised MS.
Comment: 7. In Table 3, bias value is in –ve; is there any specific reason for that.
Response: In Table 3 one value of the bias is negative (-0.08 deg C), and two values are positive (0.22 and 0.07 deg C). The values are different as they are calculated for different locations and different time periods.
Round 2
Reviewer 2 Report
Thank you authors for incorporating all the comments in the revised manuscript